# Racial and Ethnic Disparities in Chronic Stress among Male Caregivers

**DOI:** 10.3390/ijerph20126188

**Published:** 2023-06-20

**Authors:** Christine Unson, Anuli Njoku, Stanley Bernard, Martin Agbalenyo

**Affiliations:** Department of Public Health, College of Health and Human Services, Southern Connecticut State University, 493 Fitch Street, New Haven, CT 06515, USA; njokua3@southernct.edu (A.N.);

**Keywords:** caregiving, older adults, health disparities, intersectionality, caregiver burden, coping, aging, health and aging, male caregiver, masculinity, African American males

## Abstract

Whereas research on caregiving is well documented, less is known about gender inequalities in caregiver stress, coping mechanisms, and health outcomes, all of which may vary by race, ethnicity, and socioeconomic status. This scoping review investigated racial and ethnic disparities using the Stress Process Model among male caregivers. Several databases were searched including Academic Search Premier, Medline Complete, APA PsycInfo, CINHAL, Google, ProQuest, and Web of Science. Included were peer-reviewed articles in English, published from 1990 to 2022. A total of nine articles fulfilled inclusion criteria. Most of the articles indicated that compared to White male caregivers, African American male caregivers provided more hours of care, assisted with more activities of daily living (ADLs) and instrumental activities of daily living (IADLs), and experienced more financial stress. In terms of coping style, one study found African American male caregivers, compared to White male caregivers, held negative religious beliefs. Another study showed that they were at a higher risk for stroke than their White counterparts. The search revealed a dearth of studies on racial disparities in stress, coping, and health outcomes among male caregivers. Further research is needed on the experiences and perspectives of male minority caregivers.

## 1. Introduction

By 2050, almost a quarter of the population of North America will be aged 65 years or over, and the number of people over the age of 80 years will triple [1]. This sustained growth of the older adult population will increase the need for caregivers. Nearly 42 million adults in the United States provided unpaid care to an older adult in 2020 [2,3]. Women, who traditionally have been the main caregivers, make up most of the caregiver population (61%). However, as the potential supply of caregivers is dwindling [4], men are increasingly tapped for this role. The proportion of male caregivers in 2020 stood at 39% [2], up from 33% in 2009 [5]. The prevalence of male caregiving differs across racial or ethnic groups. Male caregiving is less common among African Americans (34% of all African American caregivers) compared to Whites, Hispanics, or Asians (about 40% of all caregivers) [2]. Although fewer in number, in 2020, the proportion of African American male caregivers had a five-point increase from 2009 [2,5]. Socioeconomic and health disparities are evident among caregivers of different racial or ethnic backgrounds and across genders. When compared to all caregivers, those who identify as Black or African American are more likely to be younger, have a lower income, and be unmarried [2]. In a meta-analysis of race and ethnic differences in psychological wellbeing, Liu et al. [6] showed that African American caregivers had better psychological wellbeing, but worse physical wellbeing compared to White caregivers. A meta-analysis examining gender differences among caregivers reported that compared to men, women provided a greater amount of care and had a higher prevalence of burden and depression [7]. Whereas racial, ethnic, and gender disparities in caregiving have been established in the literature [8], investigations of the intersectionality in caregiving by gender, race, and ethnicity are needed [8,9]. Studies on intersectionality in caregiving have suggested that genders by racial groups have different caregiver experiences [6] and, hence, are likely to require different types of interventions [10]. However, only a few studies have examined differences in caregiver stress for gender by racial group [6]. One barrier to such an investigation is the lack of discourse on the experiences of male caregivers, especially African American male caregivers as they pertain to their caregiver burden, coping skills, and health outcomes [9,11].

## 2. Literature Review/Theoretical Framework

### 2.1. Definition of Caregiving

Caregiving is the act of looking after the wellbeing of another person. There is informal caregiving, which is usually done by a parent, spouse, family member, or friend [12,13,14]. There is also professional caregiving, which is provided by a paid worker or organization [15]. In either case, professional or informal, Thompson [16] describes caregiving as “A complex process involving several distinct kinds of work and a great deal of work. In broad terms there is both an instrumental and emotional dimension to caregiving” [16] (p. 29). Societal norms dictate how caregiving is performed and who should perform it. In the U.S., the primary caregivers are primarily women, whereas other members of a family, including male caregivers, typically take on a secondary or ancillary role [17]. Covinsky et al. [17] defined caregiving as taking care of the activities of daily living (ADLs) or instrumental activities of daily living (IADLs) of the caregiving. Primary caregivers tend to be those who provide physical care, comfort, and assistance with ADLs rather than home management, upkeep, breadwinning, and other IADL tasks [17]. Further, when caring for an older relative, secondary caregivers tend to be the partner, spouse, or significant other of the primary caregiver. Secondary caregivers do not take on much of the physical and comfort care [18]. However, the literature shows that both primary and secondary caregivers experience physical, emotional, and financial stress [19,20,21].

### 2.2. Caregiving Stress

It is commonly acknowledged that caregiving can be stressful for the caregiver [22]. Caregiving stress can add to poor physical health and is also related to poor mental health [22]. Pearlin et al. [22,23] pointed out that the level of caregiving stress experienced correlates well with the socioeconomic conditions of the caregiver. Those with lower socioeconomic status and fewer resources to support their caregiving report greater levels of stress than those with higher education and more economic resources. Bedard and colleagues and Marino and colleagues [21,24] reported that primary caregivers tended to show more strain mentally than secondary caregivers, but both had an elevated level of emotional strain. A spouse caring for a sick spouse seems to have the highest level of stress compared to other dyads such as a parent caring for a child, elder childcaring for a parent, or someone caring for a sibling [24,25].

### 2.3. Men as Caregivers

Men who take on the role of the primary caregiver, and who take care of the care recipients’ ADLs, are many times seen as out of the ordinary because they fill a role often associated with women [25,26]. For example, fathers who actively participate in the care of their children are deemed “responsible fathers” yet the child’s mother is not given that extra label to denote her role in caregiving [27]. Other research, especially in fatherhood literature, indicates that there is a differentiation between men who follow traditional male roles of the economic provider and home manager versus a role as the primary caregiver, thus moving away from the typical hegemonic masculine role to a more “caring masculinity” [28]. These caregiver characteristics can be transferred from men who are caring for their children to men caring for adult family members and spouses.

However, most of the literature on male roles in caregiving uses a feminine lens. Friedman [29] asserted that if we look at men’s roles solely from a feminine lens, we do not open the door for male femininity but further support male hegemony and act in direct opposition of feminism. That is to say: we do not allow men to be caregivers in their own way by only expecting them to provide care in a similar fashion to female caregivers. “Some men engage in caregiving as if the responsibility was ‘work’ and they take on task-oriented care management” [16] (p. 34). Men also perform caregiving as a sense of duty. This is in keeping with the concept of hegemonic masculinity that permeates the literature on male roles and behaviors in various contexts, including caregiving [30,31,32].

Male caregivers, whether sons or partners, have been found to experience physical, emotional, and financial stress, especially when they must provide personal care and have weak social support [33]. Furthermore, Lopez-Anuarbe and Kohli [33] reported that men are less likely than women to seek out help or support in caregiving. In addition, men tend to not internalize the strains of caregiving in the same way that women do. The literature points to male caregivers having a variety of coping and stress management mechanisms that are different from women’s when it comes to caring for their own mental and physical health [34,35,36,37,38].

### 2.4. Study

Despite the body of literature on male caregiving, especially relating to responsible fatherhood, research on minority male caregiving and disparities based on the race or ethnicity of the caregiver is lacking [39]. The purpose of this scoping review was to investigate disparities in stress, coping mechanisms, and chronic health outcomes between African American and White male caregivers of older relatives. This scoping review used two theoretical models, the Stress Process Model [23] and the intersectionality framework to identify, categorize, and analyze relevant research on racial or ethnic disparities in caregiving. The results of the review could provide insight into the sources of health disparities among male caregivers of older relatives and help develop policy to decrease racial and ethnic disparities among male caregivers.

## 3. Materials and Methods

### 3.1. Identifying the Research Question

This scoping review identified and summarized research in the literature on racial and ethnic disparities in male caregiving burdens, coping styles, and health outcomes. A scoping review is a method to survey the literature by identifying key concepts, sources of information, and methods of study of a research topic [40]. The scoping review process involves five phases: (a) formulation of the research question, (b) identification of relevant studies based on established search criteria, (c) selection of relevant studies using established search criteria, (d) tabulation of the data, and (e) synthesis and reporting of the results [40].

The team searched, with the assistance of research university librarians, for relevant peer-reviewed articles with findings related to race- or ethnic-based health disparities associated with male caregiving. Several databases were searched, including Academic Search Premier, Medline Complete, APA Psycinfo, CINHAL, Google, ProQuest, and Web of Science. The original set of keywords, male and caregiving and disparities, produced no gender by race articles. Subsequently, the search was expanded to include the following keywords: (gender or sex) AND (caregiving or caregiver) AND (race or ethnicity) AND (older, elder, geriatric) AND intersectionality. We also searched by substituting male or female and Black or White for gender, sex, and race or ethnicity, respectively. We also searched by adding the keywords (burden or stress), (coping) AND (depression or depressive symptoms). For example, we used (male or female) AND (caregiving or caregiver) AND (Black or White) AND (older, elder, geriatric) AND (burden or strain). We also added (coping) OR (religion or religiosity) as well as (health outcomes) OR (depression or depressive symptoms).

### 3.2. Identifying Relevant Studies

To be included in this scoping review, a study needed to discuss male caregivers with racial or ethnic comparisons, be U.S.-based, follow the Stress Process Model [22], and focus on stressors, coping mechanisms, and health outcomes. We subsequently narrowed the search to the differences between African American male caregivers and White male caregivers when no comparisons with other ethnic groups emerged. Peer-reviewed articles from 1990 to 2022 were included in the search.

The authors reviewed about 650 articles. The articles that each author judged to have potential for inclusion were compiled in an Excel file (*n* = 144). For this initial search, the authors selected articles about gender and racial or ethnic differences in caregiving. The abstract of each article in the Excel file was reviewed independently by four team members (CU, AN, SB, and MA). This secondary review reduced the pool to 30 articles. Articles that were selected mentioned both gender and racial or ethnic differences in at least one aspect of caregiving. One of the authors, CU, extracted the specific findings in each of the 30 articles, which were then independently reviewed by two additional authors (SB and AN). Twenty-one of the thirty were eliminated because they did not provide specific differences between male African American caregivers and White caregivers. Our final sample included nine published articles (Figure 1).

## 4. Results

Nine articles met the search criteria (see Table 1). Table 1 contains the following information about the nine articles: (a) citation, (b) study design (quantitative or qualitative), (c) source of data, if quantitative, (d) sample size and composition, and relevant findings on the (e) nature of stressors or burden, (f) coping styles, and (g) health outcomes of caregivers.

Of the nine articles, seven compared caregiver strain or burden between White and African American male caregivers [41,42,43,44,45,46,47]. The remaining two articles either compared health outcomes [48] or compared coping styles [49]. All nine studies used quantitative techniques to analyze nationally or regionally representative samples. Of the nine articles, five were published in the last 5 years [41,42,44,45,49], and the remaining four [43,46,47,48] were published between 1998 and 2010.

**Table 1 ijerph-20-06188-t001:** Findings Related to Caregiver Stress or Burden, Coping Styles, and Health Outcomes of Studies that Met Inclusion Criteria.

Authors	Study Design	Data Source	Purpose	Sample	Caregiver Stress or Burden Experienced	Coping Styles and Health Outcomes
Cohen, S.A., Mendez-Luck, C.A., Greaney, M.L., Azzoli, A.B., Cook, S.K., and Sabik, N.J. (2021) [41]	Quantitative	NSOC 2011	Estimated multivariate relationship between hours of caregiving per month, number of ADLs and IADLs, and race/ethnicity, gender, and employment status.	*N* = 983 adults, 30% male. Female: White (419, 42%), Black (218, 22%), Hispanic (49, 5%). Male: White (187, 20%), Black (89, 9%), Hispanic (21, 2%)	Unemployed White males provided fewer hours per month of care and assisted with fewer IADLs than unemployed Black males. (p. 25)	Not applicable
Cohen, S.A., Sabik, N.J., Cook, S.K., Azzoli, A.B., Mendez-Luck, C.A. (2019) [42]	Quantitative	NSOC 2011	Determined if caregiving intensity as measured by hours of caregiving per month and number of ADLs and IADLs varied by race/ethnicity and gender	*N* = 993 adults, 30% male. Female: White (421, 43%), Black (220, 22%), Hispanic (49, 5%). Male: White (189, 19%), Black (93, 9%), Hispanic (21, 2%)	Black male caregivers provided significantly higher levels of ADLs, IADLs, and hours of caregiving than White male caregivers. (p. 252)	Not applicable
Fider, C.R.A., Lee, J.W., Gleason, P.C., and Jones, P. (2019) [49]	Quantitative	Biopsychosocial Religion and Health Study, a sub-study of the Advent	To determine whether religion measured prior to becoming a caregiver predicts caregiver burden, mental health, and physical health after the individual became a caregiver. (p. 1285)	559 respondents, 460 females and 124 males. Female: White (281, 50%), Black (158, 28%); Male: White (93, 17%), Black (27, 5%)	Not applicable	Black male caregivers viewed God as less loving and more controlling and had more negative interactions in church than White male caregivers did. (p. 1920)
Haley, W.E., Roth, D.L., Howard, G., and Safford, M.M. (2010) [48]	Quantitative	Reasons for Geographic and Racial Differences in STROKE (REGARDS)	Examine the effects of stress on stroke and CHD risk by race and sex.	*N* = 767 adults, 45% male. Female: White (257, 33.5%), Black (165, 21.5%). Male: White (225, 29.3%), Black (120, 15.7%)	Not applicable	Black male caregivers had the highest stroke risk score compared to other race–sex groups. (p. 333)
Laditka, J.N., and Laditka, S.B. (2001) [43]	Quantitative	Panel History of Income Dynamics, 1963	Determined gender role (sons vs. daughters), family roles (in couples or not), and race (Black vs. White) in determining whether health or non-related health help was provided.	*N* = 5458 adults, 47% male. Female: White (2134, 39%), Black (761, 14%). Male: White (2031, 37%), Black (532, 10%)	Among both daughters and sons not in couples, Blacks gave substantially more help hours than Whites did, though a much larger percentage of White sons helped compared to Black sons. (p. 443)	Not applicable
Liang, J., Aranda, M.P., Jang, Y., Wilber, K., Chi, I., and Wu, S. (2022) [44]	Quantitative	2015 National Health and Aging Trends Study (NHATS) and the National Study of Caregiving (NSOC)	Examined whether an increase in the secondary caregiver network (SCN) lessened primary caregiver burden, and whether the association varies across women and men, Black and White individuals.	*N* = 967 adults, 34% male. Female: White (402, 42.7%), Black (237, 24.5%). Male: White (226, 23.3%), Black (102, 10.5%)	Black male caregivers reported significantly lower levels of caregiver burden than White males. The increase in proportion of caregiving by SCN was associated with a faster decrease in burden among both groups of females (Black and White) than Black men caregivers. (pp. 1954–1955)	Not applicable
Liu, R., Chi, I., and Wu, S. (2022) [45]	Quantitative	NSOC 2015/2017	Explored how caregiver financial, physical, and emotional burdens are affected by gender and race/ethnicity of caregivers.	*N* = 1206 adults, 38.6% male. Female: White (369, 30.6%), Black (250, 20.8%), Other (120, 10%). Male: White (236, 19.6%), Black (139, 11.54%), Other (90, 7.5%)	Black male caregivers were most likely to report financial problems and White male caregivers were most likely to report emotional problems compared to other race or ethnic groups (p. 654)	Not applicable
Martin, C.D. (2000) [46]	Quantitative	1990 Informal Caregivers Survey (ICS)	Estimated the effects of race and gender on the relationship between a caregiver’s situation and feelings of burden.	*N* = 811, 26% male. Female: White (505, 62.2%), Black (76, 9.5%). Male: White (207, 25.5%), Black (23, 2.8%)	African American men are more likely to report feeling burdened compared to other race–sex groups (p. 1000)	
Wallsten, S.S. (2000) [47]	Quantitative	Established Populations for Epidemiological Studies of the Elderly (EOESE)	Reported the effects of caregiving, gender and race on physical health, social supports and mutuality of older couples.	*N* = 234 (caregiver–spouse couples). Non-spouse respondents: 53% male; female (White (60, 26%), Black (78, 33%)); male (White (47, 20%), Black (47, 20%)). Two (2) Native American couples were not included in the analysis.	Males and African American caregivers had more health problems and spouses with more ADLs than Caucasian and female caregivers (p. 109)	Not applicable

### 4.1. Caregiver Strain or Burden

Four of the seven studies reported that African American male caregivers were more burdened than White male caregivers because they provided higher levels of care. Based on a nationally representative sample of caregivers (NSOC), Cohen et al. [42] reported that Black male caregivers provided significantly higher levels of ADLs, IADLs, and hours of caregiving than White male caregivers. A second Cohen et al. study [41] found a similar pattern with unemployed caregivers. Unemployed African American male caregivers provided more hours of care and assisted with more IADLs than unemployed White male caregivers. African American sons were reported to provide substantially more help hours than Whites did, though more White sons provided care than Black sons. Martin [46] found that African American men were more likely to report feeling burdened compared to other race–sex groups.

African American male caregivers may experience greater burden because they have more health problems [47] and may experience more financial problems than other sex–race groups [45]. Moreover, Walsten [47] also reported that their spouses had more ADLs compared to other groups.

In contrast to these findings, Liang et al. [44] found that African American male caregivers reported a lower level of caregiver burden among the four race–gender groups. The authors suggested that African American male caregivers tended to do less hands-on care, and that they used a social caregiver network (SCN) more frequently than other groups. However, when a SCN is increased, the decrease in caregiver burden is much slower for Black men than for Black and White women.

### 4.2. Caregiver Coping Styles

Only one study mentioned a difference in the coping styles of Black men [49]. This study, which investigated the influence of religion on caregiver burden, reported that African American male caregivers perceived God as unloving and controlling and reported more negative interactions with their church than other groups, including White male caregivers. The further analysis showed that respondents who held these views experienced more caregiver burden than those who did not.

### 4.3. Caregiver Health Outcomes

Only one study examined racial and ethnic differences in health outcomes among caregivers. This study, which used a regionally representative sample to examine caregiving strain and the incidence of stroke and coronary heart disease risk among spouse caregivers, found that Black male caregivers had the highest stroke risk score compared to other race–sex groups of spouse caregivers [48]. Within the study, greater depressive symptoms were also associated with higher stroke scores.

## 5. Discussion

This scoping review identified peer-reviewed studies that reported disparities between Black and White male caregivers. We used the Stress Process Model [22,23] to categorize research findings into three groups: (a) stress or burden, (b) coping style, and (c) health outcomes. Although we found only a small number of studies, all of these studies were based on large national or regional probability samples. Moreover, most of the studies utilized objective measures of burden such as the number of ADLs and hours of work instead of a more subjective self-report. Most of the studies showed that African American male caregivers carried a heavier burden—they assisted with more ADLs and IADLs and provided more hours of care—than White male caregivers. Moreover, they were disadvantaged financially, and their care recipients required more care compared to White recipients. In terms of disparities in health outcomes, we found only one study that showed that African American caregivers were at a higher risk of stroke compared to other sex–race groups, so we cannot conclude that disparities exist in this area.

The main finding of this study, that African American male caregivers carry a heavier burden than White male caregivers, is consistent with the general finding that minority caregivers, regardless of gender, provide more intensive care and have poorer health outcomes than White caregivers. Moreover, there are aspects of male caregiving that may put this group at an additional disadvantage. As mentioned earlier, male caregivers tend not to seek support or help, express feelings of stress [16], or express discomfort with providing personal care [39], and some of them may endure until their burdens become overwhelming [39]. This will make their jobs of caregiving more difficult. Men should be encouraged to seek help regularly to promote their physical and socioemotional wellbeing. This encourages a need for educational programs that promote awareness of caregiving services and supports (e.g., local and federal policies to support family caregivers) among male caregivers, but especially African American caregivers. However, Schwarz and McInnis-Dittrich [50] noted that support services are often not designed to reach out to male clients who tend to be wary of formal and informal social supports and view help-seeking as a personal weakness.

This study found that African American males were disinclined to view their God as caring and had more negative interactions with their church. This finding, though based on only one study, suggests that religiosity may not be a workable coping mechanism for African American men. Kazemi et al. [51] reported that male caregivers of stroke patients used the skills of positive reappraisal and accepting responsibility more than female caregivers and recommended training programs that increase the adoption of effective and healthy coping strategies.

One important finding of this scoping review is the lack of studies on racial disparities in health outcomes of male caregivers. In this study, spousal African American male caregivers had the highest level of stress, which was, in turn, highly associated with risk for stroke. Chronic stress may result, among other reasons, from having to care for someone with behavioral problems or watching someone experiencing pain [34,48]. More studies on health outcomes may help resolve the paradox in which minority caregivers provide more intensive care and yet report less stress and fewer emotional difficulties [9,39].

Overall, little is known about male caregiving relative to female caregiving because historically male caregivers, especially minority male caregivers, were much smaller in number than female caregivers. The American Association of Retired Persons (AARP), however, reported that the proportion of male caregivers to total caregivers is changing and increased to as much as 40% of all caregivers in 2017 [52]. This trend suggests that there should be more studies on male caregiving commensurate to the growing presence of male caregivers.

### Application of Theoretical Frameworks

This review applied the Stress Process Model to categorize existing studies on racial and ethnic disparities between African American and White male caregivers. In the Stress Process Model [22,23], the three main components are (a) primary (e.g., problem behaviors) and secondary (e.g., financial problems) role strains, which lead to intrapsychic strain (e.g., loss of self-esteem) and negative health outcomes (e.g., depression); (b) mediators such as coping skills and social support that can attenuate the effects of the stressors on intrapsychic strain; and (c) health outcomes.

We also applied the intersectionality framework by investigating African American and White differences in male caregiving. The intersectionality framework posits that individual sociodemographic characteristics (i.e., gender, race or ethnicity, socioeconomic status, age and geographic location) interact to produce a unique life experience that is shaped by access to resources and opportunities and the institutional forces of privilege and oppression (e.g., racism, sexism and ageism) [53]. Membership in more than one of these groups increases the potential for caregiver disparity and inequity.

Given the small number of studies identified, we were unable to consider factors other than gender and race. Dilworth-Anderson et al. [54] explained that factors influencing caregiving are “nuanced”, as they may not be just interacting with one other factor; multiple factors may be at play. The Sociocultural Stress and Coping Model [54] posits that cultural values, such as familism and religiosity, affect how stress is appraised and coping strategies are adopted. Additionally, previous studies have shown that cultural aspects may allow African American male caregivers to better cope with caregiving, including spiritualty, a deep sense of commitment, a supportive network, and a desire to keep the care recipient in the home rather than in a nursing home [39,55]. Essentially, community and culture can help African American caregivers manage the challenges of family caregiving [56]. Therefore, future interventions to reduce disparities should be cognizant of such cultural attributes of caregiving.

## 6. Recommendations

This review found that disparities pertaining to burdens of caregiving, coping styles, and health outcomes between racial or ethnic groups, particularly between African Americans and Whites, are understudied. Because of this, our first recommendation is for much-needed research on the risk factors leading to disparities in order to tailor interventions to male caregivers from specific racial/ethnic and socioeconomic groups. We recommend two theoretical frameworks to guide future research on health disparities. The first is an intersectionality framework, which requires researchers to examine how multiple identities compound the effects of racism, sexism, and other forms of inequities on caregivers’ health and wellbeing [53]. Results may shed light on the differential needs of specific groups. For example, some male groups may need sustainable safety net programs (e.g., assistance to unpaid male caregivers in financial need) [33], whereas other groups, such as those reluctant to seek help or whose main needs are informational, would benefit from counseling or educational outreach services or technology to access information.

A second theoretical framework that should guide research in male caregiving is the “caring masculinity” framework, which contradicts hegemonic masculinity. The latter is embodied in the attitudes and practices that men use to dominate women and other men, thereby perpetuating gender and race–ethnic inequality [57]. The former perceives men as capable of expressing emotions, working interdependently, and having meaningful relationships with others involved in caregiving [58]. Alongside this perspective, researchers must also expand their definition of caregiving to include tasks men consider as caregiving [59]. A gender-neutral view of caregiving could lead to the design of interventions, such as supportive services, that meet the needs of all caregivers.

### Study Limitations

This study is not without limitations. First, the fact that only a few studies met the inclusion criteria may be due to a faulty search strategy. Additionally, this scoping review, given the small number of studies reviewed, did not differentiate between caregiving provided by a spouse, a relative, or an adult child. Those caring for an adult child tend to experience a lower quality of life and more caregiver burden [59]. Neither did this study differentiate caregiving provided in the context of dementia or other health conditions. Dementia care has been reported as the most burdensome compared to other health conditions [55]. Third, we only looked for statistically significant relationships and did not report relationships that were not significant when these were available. The possibility that non-significant relationships were not reported by research studies is also recognized. In addition, a lack of significant differences may stem from the relatively small number of minority male caregivers in the datasets. Fourth, we only examined studies that were published in English. Finally, we did not examine studies in which caregivers provided both elder care and care for children or grandchildren (i.e., multi-generational care). This review’s strength is its intersectional exploration of male caregiving contexts and health outcomes by race or ethnicity.

## 7. Conclusions

The prevalence of caregiving has grown, and men are increasingly assuming roles as family caregivers. This scoping review examined racial and ethnic disparities in chronic stress, coping styles, and health outcomes between White and African American male caregivers. A small number of studies showed that African American male caregivers were disadvantaged financially, provided more hours of care, assisted with more ADLs and IADLs, were less religious, and were at a higher risk for stroke than White male caregivers. The results showed a lack of studies on the health disparities between White and African American male caregivers. Because of this, we recommend the adaptation of an intersectionality framework to study the cumulative effects of inequities associated with demographic characteristics on caregiver burden and health outcomes; it is important for future research to develop interventions tailored to the needs of these intersectional groups.

## Figures and Tables

**Figure 1 ijerph-20-06188-f001:**
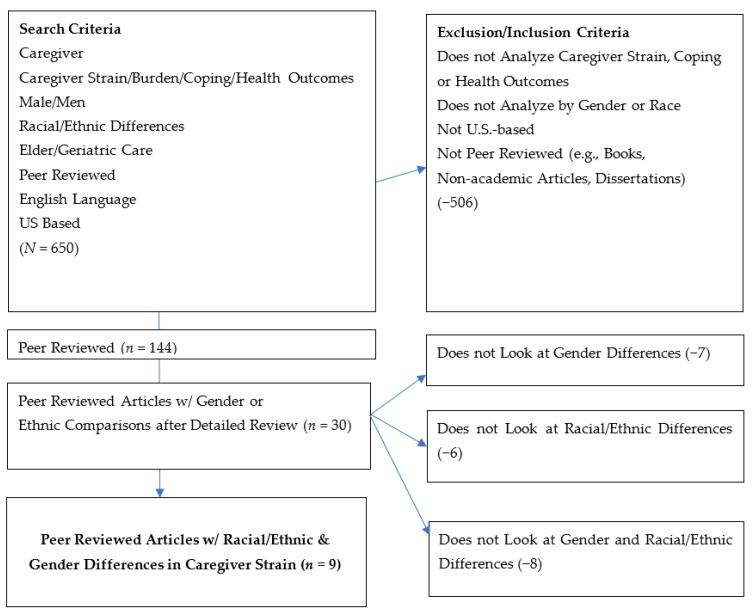
Inclusion and Exclusion Criteria Scoping Review.

## Data Availability

Not applicable.

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
