# Peer review of "Racial and Ethnic Disparities in Chronic Stress among Male Caregivers"

_ijerph, 2023, doi:10.3390/ijerph20126188_

Round 1

Reviewer 1 Report

Overall, the study has potential to add to the literature, but it needs significant editing, reorganization, and attention to detail. In addition, some of oversights in key areas also made me wonder if other important data is missing.

Major Comments:

Significant attention needs to be paid to how the study is written and organized. For example, the introduction is a list of statistics, many of which are repetitive and it’s difficult to follow. These two paragraphs could be pared down significantly to focus on the main points. This is true for the majority of the manuscript. Most of the sections have redundant sentences or paragraphs.

Methods are not sufficiently detailed. That is, they are not written in a manner that one could easily duplicate. Typically, a study like this would have a detailed appendix with sample searches etc. I am also curious as to why the word “disparities” wasn’t included in the search, as that is the main goal of the paper – assessing literature on disparities.

Research question is not clear. Until I read the “results” table, I thought the study was only on minority male caregivers. In fact, it’s broader and compares minority male caregivers with both white male caregivers and female caregivers of all races.

The inclusion of theoretical frameworks makes sense, but it is not well developed. Initially the authors indicate they will use an intersectionality framework, but the section on theory mentions several different theories. Further, the authors take for granted that readers are familiar with all of the frameworks they discuss. If the study is going to be organized around a theory, it should be explained clearly and in detail, as should how it will be used to understand the literature they review.

Minor Comments:

 IADLs are instrumental activities of daily living

The Laditka et al study used the Panel Study of Income Dynamics from 1993Overall, the study has potential to add to the literature, but it needs significant editing, reorganization, and attention to detail. In addition, some of oversights in key areas also made me wonder if other important data is missing.

Major Comments:

·       Significant attention needs to be paid to how the study is written and organized. For example, the introduction is a list of statistics, many of which are repetitive and it’s difficult to follow. These two paragraphs could be pared down significantly to focus on the main points. This is true for the majority of the manuscript. Most of the sections have redundant sentences or paragraphs.

·       Methods are not sufficiently detailed. That is, they are not written in a manner that one could easily duplicate. Typically, a study like this would have a detailed appendix with sample searches etc. I am also curious as to why the word “disparities” wasn’t included in the search, as that is the main goal of the paper – assessing literature on disparities.

·       Research question is not clear. Until I read the “results” table, I thought the study was only on minority male caregivers. In fact, it’s broader and compares minority male caregivers with both white male caregivers and female caregivers of all races.

·       The inclusion of theoretical frameworks makes sense, but it is not well developed. Initially the authors indicate they will use an intersectionality framework, but the section on theory mentions several different theories. Further, the authors take for granted that readers are familiar with all of the frameworks they discuss. If the study is going to be organized around a theory, it should be explained clearly and in detail, as should how it will be used to understand the literature they review.

Minor Comments:

·       IADLs are instrumental activities of daily living

·       The Laditka et al study used the Panel Study of Income Dynamics from 1993 

Author Response

May 19, 2023

Reviewer 1

Dear Sir or Madam,

First of all, we would like to thank you for the opportunity to revise and resubmit our manuscript. We are very grateful for your detailed and critical comments. We have attempted to fully comply with your comments which have greatly improved the quality of our manuscript.

We apologize for the delay in submission of the revisions.

The discussion below addresses your major and minor reviewer comments.

Major Comments:

  • Significant attention needs to be paid to how the study is written and organized. For example, the introduction is a list of statistics, many of which are repetitive and it’s difficult to follow. These two paragraphs could be pared down significantly to focus on the main points. This is true for the majority of the manuscript. Most of the sections have redundant sentences or paragraphs.

Our response: We radically revised the introduction by removing statistics unrelated to the US, and unrelated to male caregiving. We presented statistics that focused on the need for male caregivers, and the rise in the number of male caregivers, thereby laying the groundwork for the rationale of the study. (See lines 35 to 43)

We also revised the second paragraph by stating the main point as the first sentence of the paragraph and removing the statistical information. We also strengthened our arguments by citing meta-analysis studies instead of individual studies and provided statements that hinted the study’s intent and method. (See lines 47-63)

By 2050, almost a quarter of the population of North America will be age 65 and over and number of people over the age of 80 will triple [1]. This sustained growth of the older adult population will increase the need for caregivers. Nearly one in five or about 53 million adults in the United States provided unpaid care to an older adult in 2020 [2]. Women, who have been traditionally the main caregivers, make up the majority of the caregivers (61%). However, as the potential supply of caregivers is dwindling [3], men are increasingly being tapped to be caregivers. The proportion of male caregivers in 2020 stands at 39% [2], up from 33% in 2009 [5]. The prevalence of male caregiving differs across race or ethnic groups. Male caregiving is less common among African Americans (34% of all African American caregivers) compared to whites, Hispanics or Asian (about 40% of all caregivers) [2]. Although fewer in number, the proportion of male African American caregivers is a five-point increase from 2009 [5].

Socio-economic and health disparities are evident among caregivers of different race or ethnic background and across gender. When compared to all caregivers, those who identify as Black or African American are more likely to be younger, have lower income, and unmarried [2]. Liu et al (2021) in their meta-analysis of race and ethnic differences in psychological well-being showed that African American caregivers had better psychological well-being but worse physical well-being compared to white caregivers ([6] p. e232). A meta-analysis examining gender differences among caregivers reported that compared to men, women provided a greater amount of care [2] had a higher prevalence of burden and depression [7], p. 38]. Whereas racial, ethnic, and gender disparities in caregiving have been established in the literature [8], investigations of the intersectionality in caregiving by gender, race and ethnicity are needed [8, 9] Studies on intersectionality in caregiving have suggested that gender by race groups have different caregiver experiences [6] and hence, are likely to require different types of interventions [10]. However, only a few studies have examined differences in caregiver stress in gender by race groups [6]. One barrier to such an investigation is the lack of discourse on the experiences of male caregivers, especially African American male caregivers as it pertains to their caregiver burden, coping skills and health outcomes [11,12].

  • Methods are not sufficiently detailed. That is, they are not written in a manner that one could easily duplicate. Typically, a study like this would have a detailed appendix with sample searches etc.

Our response: We conducted additional search after receiving comments from the reviewers. The paragraphs describing the process are shown below. The yellow highlights are the changes we made.

The team searched with the assistance of research university librarians, for relevant peer-reviewed articles that had findings related to race or ethnic-based health disparities associated with male caregiving. Several databases were searched including Academic Search Premier, Medline Complete, APA Psycinfo, CINHAL, Google, ProQuest, and Web of Science. The original set of keywords: male and caregiving and disparities produced no gender by race articles. Subsequently, the keyword search was expanded to include the following keywords: (gender or sex) AND (caregiving or caregiver) AND (race or ethnicity) AND (older, elder, geriatric) AND intersectionality.  We also searched by substituting male or female and black or white for gender and sex and race or ethnicity, respectively. We also searched by adding the keywords (burden or stress), (coping) and (depression or depressive symptoms). For example: we used (male or female) AND (caregiving or caregiver) AND (Black or White) AND (older, elder, geriatric) AND (burden or strain). We substitute (coping) or (health outcomes). We also used (religion or religiosity) and (depression or depressive symptoms). The supplemental search produced one additional article.

Note: In the original version of the manuscript, we had nine articles, of which one was a qualitative study. The qualitative study was about differences in coping styles by males from different race and ethnic groups. We decided to remove the article from the list because we could not confidently draw a conclusion from one qualitative study.

3.2. Identifying Relevant Studies

To be included in this scoping review, a study must present gender or sex by race or ethnic comparisons, must be U.S.-based studies, and following the Stress Process Model [17], focused on stressors, coping mechanisms and health outcomes of male caregivers in comparison with male caregivers of other ethnic groups. We subsequently narrowed the search to male African American in comparison to male white caregivers when no comparisons with other ethnic groups emerge.  Peer-reviewed articles from 1990 to the present were included in the search. 

The authors reviewed about 650 articles. The articles that each author judged to have potential for inclusion were compiled in an Excel file (n = 144). For this initial search, the authors selected articles about male caregiving, gender and race or ethnic differences in caregiving. The abstract of each article in the file was reviewed independently by four team members (CU, AN, SB, MA). This secondary review reduced the number to 30 articles. Articles that were selected mentioned both gender and race or ethnic differences in at least one aspect of caregiving. One of the authors (CU), extracted the specific findings in each of the 30 articles which were then independently reviewed by two of the four authors (SB, AN). The final nine were accepted for review in this study. Twenty-one of the 30 were eliminated because they did not provide specific differences between African American male caregivers and white male caregivers. Our final sample included nine published articles (Figure 1).

  • Typically, a study like this would have a detailed appendix with sample searches etc. 

Our response: We have an excel file showing the articles we downloaded and how we disposed each of the article.  However, we believe that the flow chart showing the disposition of the articles provides the same information.

4) I am also curious as to why the word “disparities” wasn’t included in the search, as that is the main goal of the paper – assessing literature on disparities. 

Our response: As mentioned in the above paragraph (highlighted in yellow), our initial search using disparities as a keyword was not productive.

  • Research question is not clear. Until I read the “results” table, I thought the study was only on minority male caregivers. In fact, it’s broader and compares minority male caregivers with both white male caregivers and female caregivers of all races.

The study IS on male caregivers – African American and White. We have added several devices to make the focus clearer: a) we changed the title of the manuscript to better reflect the topic, b) the introduction and literature review hints at socio-economic disparities and male caregiving, and c) the study (See 2.4 The Study, lines 133-143), specifically states the purpose of the study.

2.4  Study

Despite a body of literature on male caregiving especially with responsible fatherhood, research on minority male caregiving and disparities based on the race or ethnicity of the caregiver is underrepresented [40]. The purpose of this scoping review is to investigate disparities, in stress, coping mechanisms and chronic health outcomes between male African American and white caregivers of older relatives. This scoping review study used two theoretical models, the Stress Process Model [24], and the intersectionality framework to identify, categorize, and analyze relevant research on race or ethnic disparities in caregiving. The results of the study could provide insight into the sources of health disparities among male care givers of older relatives and help develop policy to decrease racial and ethnic disparities in chronic stress among male caregivers.

  • The inclusion of theoretical frameworks makes sense, but it is not well developed. Initially the authors indicate they will use an intersectionality framework, but the section on theory mentions several different theories. Further, the authors take for granted that readers are familiar with all of the frameworks they discuss. If the study is going to be organized around a theory, it should be explained clearly and in detail, as should how it will be used to understand the literature they review.

Our response: We have revised the Theoretical Framework section radically. We explained the Stress Process Model, Intersectionality Framework and Sociocultural Stress and Coping Model and explained how we used these theories in the analysis. We deleted reference to the other models we cited. See below for the revised section on Theoretical Framework (Lines 298-328).

5.1. Application of Theoretical frameworks

This study applied the Stress Process Model to categorize the studies on race and ethnic disparities between African American and white male caregivers into three main components of the model. In the Stress Process Model [23, p. 586], the main components are a) primary (e.g., problem behaviors) and secondary role strains (e.g., financial problems) which lead to intrapsychic strain (e.g., loss of self-esteem) and negative health outcomes (e.g., depression). However, b) mediators such as coping skills and social support can attenuate the effects of the stressors on intrapsychic strain, and c) health outcomes. We also applied the intersectionality framework by investigating African American and White differences in male caregiving. The intentionality framework posits that individual socio-demographic characteristics (i.e., gender, race or ethnicity, socio-economic status, age, geographic location) interact to produce a unique life experience that is shaped by access to resources and opportunities and the institutional forces of privilege and oppression [e.g., racism, sexism, ageism [54]. Membership in more than one of these groups increases the potential for stress. 

Given the small number of relevant studies we identified, we were unable to consider factors other than gender and race as the intersectionality framework prescribes and other models suggest. Dilworth-Anderson et al. (2005) explained that factors influencing caregiving are “nuanced” as they may not be just interacting with one other factor; multiple factors may be at play [55].  The Sociocultural Stress and Coping Model [55] posits that culture values such as familism and religiosity affect how stress is appraised and which coping strategies to adopt. Additionally, previous studies have shown that cultural aspects may allow African American male caregivers to better cope with caregiving, including spiritualty, a deep sense of commitment, a supportive network, and a desire to keep the care recipient in the home rather than in a nursing home [40, 56]. Essentially, community and culture can help Black caregivers manage the challenges of family caregiving [57]. Therefore, interventions to reduce disparities should be designed that are cognizant of such cultural attributes of caregiving.  We were also unable to apply the socio-economic aspects of intersectionality. conditions between African American and white male caregivers. Studies have shown that African American male socio-economic conditions that are associated with ethnicity are a major cause of ethnic disparity [2]

  • Most of the sections have redundant sentences or paragraphs.

Our response: We revised the Introduction, Application of Theoretical Frameworks and the Recommendation Sections where we believe contained the most redundancy.

We radically pared down the Recommendation Section. We limited the recommendations to research recommendations, given the lack of research in male caregiving, and in disparities in male caregiving. Recommendations to uplift the well-being of male caregivers were kept to a minimum, given the small number of studies uncovered by this study. See below: (lines 337-360)

  1. Recommendations

This study found that disparities pertaining to burdens of caregiving, coping styles and health outcomes between race or ethnic groups, particularly between African Americans and whites, are understudied. Hence, our first recommendation is for much needed research on the risk factors leading to disparities in various aspects of male caregiving in order to identify causes of disparities and to tailor interventions to male caregivers from specific race/ethnic, and socio-economic groups. We recommend two theoretical frameworks to guide future research on health disparities. The first is intersectionality framework which requires researchers to examine how multiple identities compound the effects racism, sexism, and other forms of inequities on the health and well-being of a caregiver [58]. Results may shed light on differential needs of specific groups. For example, some male groups may need sustainable safety net programs (e.g., assistance to unpaid male caregivers in financial need) [33] whereas other groups  - such those reluctant to seek help or whose main needs are informational – would benefit from counseling or educational outreach services or technology to access information.

A second theoretical framework that should guide research in male caregiving is the “caring masculinity” framework which contradicts “hegemonic masculinity.” The latter is embodied in the attitudes and practices that men use to dominate women and other (e.g., racial minority) men thereby perpetuating gender and race-ethnic inequality [59, p. S113].  The former framework perceives men as capable of expressing emotions, working interdependently and having meaningful relationships with others involved in caregiving [60] Alongside this perspective, researchers must also expand their definition of the scope of caregiving to include tasks men consider as caregiving [61]. A gender-neutral view of caregiving could lead to the design of interventions such as supportive services that meet the needs of both male and female caregivers.

Minor Comments:

  • IADLs are instrumental activities of daily living

Our response. We have made the change. Thank you for catching this.

  • The Laditka et al study used the Panel Study of Income Dynamics from 1993.

Our response: We have made the change. Thank you for catching this.

  • Overall, the study has potential to add to the literature, but it needs significant editing, reorganization, and attention to detail.

In addition, some of oversights in key areas also made me wonder if other important data is missing.

Our response: The manuscript has been gone through a major revision. We hope that the changes have been satisfactory.

We hope that our responses satisfy your requirements. Once again, we thank you for your insightful comments and for the opportunity to revise and resubmit our manuscript. We apologise for the late submission.

Sincerely,

Christine Unson PhD

Anuli Njoku PhD

Stanley Bernard PhD

Martin Agbalenyo MPH

Reviewer 2 Report

Dear authors,

please see the attached file with my commends and recommendations regarding your manuscript.

Author Response

May 19, 2023

Reviewer 2

Dear Sir or Madam,

First of all, we would like to thank you very much for the opportunity to revise and resubmit our manuscript. We are deeply grateful for your detailed and critical comments. We have attempted to fully comply with your comments which we believe have greatly improved the quality of our manuscript.

We apologize for the delay in submission of the revisions.

The discussion below addresses your major and minor comments.

  1. ABSTRACT
  • lines 20-21: “Peer-reviewed articles in English that were published from 1990 to 2022 were included.” Please specify whether the same time range (1990-2022) was applied to other material and research, besides peer-reviewed articles.

Our response: We only searched for peer-reviewed articles.

  1. INTRODUCTION
  • lines 36-38: It is not clear how racial and ethnic diversity in the U.S. is related with the need to ensure that individuals grow old healthily. Please clarify.

Our response: The sentence has been eliminated.

  • lines 39-40: “However, the majority of 39 caregivers are unpaid family members”. Authors, please provide a reference for this statement.

Our response: The sentence has been eliminated.

  • line 51: Please clarify the notion “caregiving intensity”.

Our response: The phrase has been eliminated.

  • Lines 56-60: “The purpose of this scoping.... among minority male caregivers”. These sentences need to be removed at the end of section 2.3, on page 3, and merge with those at lines 125-129.

Our response: We created a section “2.4 The Study”. See Section 1.4 (The Study) and lines 129-138 of the final manuscript at the end of the literature review. We removed lines 56-60 and incorporated it in the section “2.4 The Study.” (See Lines 132-142).

      2.4  The Study

Despite a body of literature on male caregiving especially with responsible fatherhood, research on minority male caregiving and disparities based on the race or ethnicity of the caregiver is underrepresented [40]. The purpose of this scoping review is to investigate disparities, in stress, coping mechanisms and chronic health outcomes between African American and white male caregivers of older relatives. This scoping review study used two theoretical models, the Stress Process Model [24], and the intersectionality framework to identify, categorize, and analyze relevant research on race or ethnic disparities in caregiving. The results of the study could provide insight into the sources of health disparities among male care givers of older relatives and help develop policy to decrease racial and ethnic disparities in chronic stress among male caregivers.

  • The sentence: “the majority of caregivers are unpaid caregivers” has to be supported with a cite.

Our response: See line 37 “Nearly 42 million adults in the United States provided unpaid care to an older adult in 2020 [2].”

  1. LITERATURE REVIEW
  • lines 64-69: Authors must additionally provide 1-2 references regarding professional and informal caregiving. 

Our response: We have provided additional reference:

There is informal caregiving usually done by a parent, spouse, family member or friend [13, 14, 15]. There is also professional caregiving, which is provided by a paid worker or organization [16]. In any case, professional or informal, [17 ] describes caregiving as “A complex process involving several distinct kinds of work and a great deal of work. In broad terms there is both and instrumental and emotional dimension to caregiving” (p. 29). Lines 67-70.

  • lines 70-76: “Societal norms dictate..... home management, upkeep, and breadwinning”. This is a “rich of meanings” paragraph (i.e. societal norms, primary caregivers, women caregivers in western culture etc), but no documentation is provided! Authors must provide a sufficient number of references to support their claims.

Our response:  We have deleted the aforementioned phrase, simplified the statement and provided a reference. See Lines 72-75.

Societal norms dictate how caregiving is performed and who should be the performer. In the U.S., the primary caregivers are primarily women, whereas other members of a family, including male caregivers, typically take on a secondary or ancillary role to women caregivers [18].

  • line 99-100: “....tend to be seen in western culture as “extra-ordinary” men.”. Please provide a reference for this claim. 

Our response: We have deleted the aforementioned phrase, tempered the statement and provided a reference. See Lines 101 -103.

Men who take on the role of primary caregiver, taking care of the care recipients’ activities of daily living (ADL), are many times seen as being out of the ordinary because they fill a role often associated with women [26, 27].

  • lines 107-109: “These caregiver superlatives.... family members and spouses.” This is sentence with an unclear meaning. Please clarify.

Our response:  We changed the wording. See Lines 109-110.

These caregiver characteristics can be transferred from men who are caring for their children to men caring for adult family members and spouses.

  • line 111: “......that if we continue to look at women’s role....”: Do you mean “men’s role”? 

Our response: Yes, thank you for the observation.

  • lines 122-124: “Men can compartmentalize and..... the way women do”: Unclear sentence. Please rephrase. 

Our response: We revised the statement to read:

In addition, men tend to not internalize the strains of caregiving in the same way that women do. The literature points to male caregivers having a variety of coping and stress management mechanisms that are different from women’s when it comes to caring for their mental and physical health [35-39]. See Lines 123-126.

  • Line 125: “Despite a body of literature on male caregiving stress...” Surprisingly, male caregiving stress is not reported in this section (2.3)! One or two sentences with references must be added at the beginning of the previous paragraph (line 120). 

Our response: See the paragraph below (lines 123 to 130)

Male caregivers, whether sons or partners have been found to experience physical, emotional and financial stress, especially when they have to provide personal care and have weak social support [34]. Furthermore, Lopez-Anuarbe and Kohli reported that men are less likely than women to seek out help or support to take on the task of caregiving [34]. In addition, men tend to not internalize the strains of caregiving in the same way that women do. The literature points to male caregivers having a variety of coping and stress management mechanisms that are different from women’s when it comes to caring for their mental and physical health [35-39].

  1. METHODS

  • lines 144-148: Did authors used the terms “stress” AND/OR “health outcomes” AND/OR “coping” in the search algorithm? These are core terms for the review!! Luck of these terms may have jeopardized the findings!

Our response: Following your comment, we conducted additional searches using those terms: See below:

For example: “we used (male or female) AND (caregiving or caregiver) AND (Black or White) AND (older, elder, geriatric) AND (burden or strain). We substitute (coping) or (health outcomes). We also used (religion or religiosity) and (depression or depressive symptoms). The supplemental search produced one article.” (Lines 157-167)

  • lines 154-155: “.....and health outcomes of male caregivers in comparison with male caregivers of other ethnic groups.” It is not clear in which characteristic of male caregivers were compared with male caregivers of other ethnic groups (i.e. native Americans vs all other ethnic groups?). Please clarify.

Our response: We revised the sentence to read:  

“We subsequently narrowed the search to differences between male African American caregivers and male white caregivers when no comparisons with other ethnic groups emerged.’  (Lines 168-169)

  • Lines 157-164: This paragraph needs further clarifications.
  1. On which basis the four authors initially judged and ended up to the 132 articles?  

Our response: “For this initial search, the authors selected articles about male caregiving, gender and race or ethnic differences in caregiving. The abstract of each article in the file was reviewed independently by four team members (CU, AN, SB, MA). This secondary review reduced the number to 30 articles. Articles that were selected mentioned both gender and race or ethnic differences in at least one aspect of caregiving.” (Lines 172 -177).

  • Who were the two team members who independently reviewed the 132 articles? 

Our response: The abstract of each article in the file was reviewed independently by two team members (CU, AN) 

  • Which criteria they applied to end up to the 18 articles? 

Our response: This secondary review reduced the number to 30 articles. Articles that were selected mentioned both gender and race and ethnicity differences in at least one aspect of caregiving. 

  • Please change the order of the last three sentences of the paragraph so that they reflect the sequence: you first excluded nine articles and then the remaining nine were included in the review.

Our response: “The final nine were accepted for review in this study. Twenty-one of the 30 were eliminated because they did not provide specific differences between African American male caregivers and white male caregivers.” (Lines 179-180).

  • lines 165-166: Please specify (give the initials) which of the authors initially extracted the

 information of the Table 1 and who were the other two who confirmed the extraction. 

Our response:  One of the authors conducted the initial data extraction (CU) and two

authors (AN, SB) independently confirmed or disconfirmed the extracted information.

  1. RESULTS
  • lines 173-179: Please provide the citations of the selected studies that you describe in this paragraph (i.e. “Of the 9 articles, seven articles compared....”). Which are these seven articles? Otherwise it is of no meaning the two citations [35, 36] you report in this paragraph.

Our response:

We found nine articles that met the search criteria (See Table 1). Of the 9 articles, seven articles compared caregiver strain or burden across gender by ethnic groups [42-48].  The remaining two articles either compared health outcomes [49] or compared coping styles [50]. All nine studies used quantitative techniques to analyze nationally or regionally representative samples. Of the nine articles, five were published in the last 5 years [42-46, 50] and the remaining were published between 1998 and 2010.  See Lines 192-197.

Note : In the original version of the manuscript, we had nine articles, of which one was a qualitative study. The qualitative study was about differences in coping styles by males from different race and ethnic groups. We decided to remove the article from the list because we could not confidently draw a conclusion from one qualitative study with a small sample.

  • Table 1. It is not clear what exactly is described in column titled “Stress or Burden” and why it is empty in studies #1,4?

Our response: We relabeled column heading titled “Stress or Burden” to “Findings about Caregiver Strain or Burden.”  The cells of #1 and #4 studies are empty because those studies focused on coping and health outcomes. We also added not applicable in the empty cells.

  • Similarly, if no health outcomes were reported in studies #2,3,5,7,8,9 why these were included in the review?

Our response: There are empty cells with #2,3,5,7,8,9 under health outcomes because these studies focused on caregiver strain or burden.

  • Finally, burden found on the study of Liang et al (2022) is reported by authors at the “Coping style” column of the Table. It is confusing.

Our response: We have made the correction. Thank you for the observation.

  1. DISCUSSION
  • lines 243-244: Authors must wonder whether their conclusion that based on only one study is valid.

Our response: See line 256-259 (partly shown below)

In terms of disparities in health outcomes, because we found only one study that showed that African-American caregivers were at higher risk of stroke compared to other sex-race groups, we cannot conclude that disparities exist in this area. 

  • Line 249: What does AARP stand for?

Our response: see phrase in red.

. The American Association of Retired Persons (AARP), however, reported that the proportion of male caregivers to total caregivers is changing and has increased to as much as 40% of all caregivers in 2017 [3]. See line 291     

  • Lines 254-255: Authors must consider the possibility the lack of studies to be due to problematic search strategy. This can be discussed.

Our response: See sentence in red in the study limitations section (6.1) which we included.

This study is not without limitations. First, that only a few studies that met criteria were found may be due to a faulty search strategy. (Line 359)

  • Line 258: Delete duplicated reference

Our response: We have deleted in the duplicated reference.

  • Lines 264-266: It is not at all clear to me how the theoretical models reported here are highlighted in this review! Except the intersectionality theory, no model is previously mentioned and explained how it was used in the search strategy. Stress and coping is not at all discussed by authors – let alone in the light of the reported theories!

Our response: We have revised the last paragraph of the literature review and theoretical framework section to include a description of the Stress Process Model which includes stress, coping and health outcomes, shown below:

2.4  Study

Despite a body of literature on male caregiving especially with responsible fatherhood, research on minority male caregiving and disparities based on the race or ethnicity of the caregiver is underrepresented [40]. The purpose of this scoping review is to investigate disparities, in stress, coping mechanisms and chronic health outcomes between male African American and white caregivers of older relatives. This scoping review study used two theoretical models, the Stress Process Model [24], and the intersectionality framework to identify, categorize, and analyze relevant research on race or ethnic disparities in caregiving. The results of the study could provide insight into the sources of health disparities among male care givers of older relatives and help develop policy to decrease racial and ethnic disparities among male caregivers.

Our response: We have revised the Theoretical Framework section radically. We explained the Stress Process Model, Intersectionality Framework and Sociocultural Stress and Coping Model and explained how we used these theories in the analysis. We deleted reference to the other models we cited. See below for the revised section on Theoretical Framework (Lines 298-328).

5.1. Application of Theoretical frameworks

This study applied the Stress Process Model to categorize the studies on race and ethnic disparities between African American and white male caregivers into three main components of the model. In the Stress Process Model [23, p. 586], the main components are a) primary (e.g., problem behaviors) and secondary role strains (e.g., financial problems) which lead to intrapsychic strain (e.g., loss of self-esteem) and negative health outcomes (e.g., depression). However, b) mediators such as coping skills and social support can attenuate the effects of the stressors on intrapsychic strain, and c) health outcomes. We also applied the intersectionality framework by investigating African American and white differences in male caregiving. The intentionality framework posits that individual socio-demographic characteristics (i.e., gender, race or ethnicity, socio-economic status, age, geographic location) interact to produce a unique life experience that is shaped by access to resources and opportunities and the institutional forces of privilege and oppression [e.g., racism, sexism, ageism [54]. Membership in more than one of these groups increases the potential for stress. 

Given the small number of relevant studies we identified, we were unable to consider factors other than gender and race as the intersectionality framework prescribes and other models suggest. Dilworth-Anderson et al. (2005) explained that factors influencing caregiving are “nuanced” as they may not be just interacting with one other factor; multiple factors may be at play [55].  The Sociocultural Stress and Coping Model [55] posits that culture values such as familism and religiosity affect how stress is appraised and which coping strategies to adopt. Additionally, previous studies have shown that cultural aspects may allow African American male caregivers to better cope with caregiving, including spiritualty, a deep sense of commitment, a supportive network, and a desire to keep the care recipient in the home rather than in a nursing home [40, 56]. Essentially, community and culture can help Black caregivers manage the challenges of family caregiving [57]. Therefore, interventions to reduce disparities should be designed that are cognizant of such cultural attributes of caregiving.  We were also unable to apply the socio-economic aspects of intersectionality. conditions between African American and white male caregivers. Studies have shown that African American male socio-economic conditions that are associated with ethnicity are a major cause of ethnic disparity [2]

Our response: We have included in the literature review, a paragraph on male stress and coping (lines 123-130)

Male caregivers, whether sons or partners have been found to experience physical, emotional and financial stress, especially when they have to provide personal care and have weak social support [34]. Furthermore, Lopez-Anuarbe and Kohli reported that men are less likely than women to seek out help or support to take on the task of caregiving [34]. In addition, men tend to not internalize the strains of caregiving in the same way that women do. The literature points to male caregivers having a variety of coping and stress management mechanisms that are different from women’s when it comes to caring for their mental and physical health [35-39].

  • Line 273: Why this earlier work was not included in the review? If the particular study did not fit in the inclusion criteria why authors make this expect statement?

Our response: The Dilworth-Anderson study did not have gender by race comparisons.

  • Overall, the discussion of one particular study [43] in such a detail (lines 271-283] seems meaningless – at least unconnected with the findings.

Our response: The discussion was deleted.

  • Lines 326-335: The paragraph regarding COVID-19 and the related practices does not seem to fit with the results and discussion of the current review. How (non) vaccination is related to any of the issues addressed in this review (i.e. stress, burden, health outcomes)?

Our response:  The paragraph has been deleted.

  1. CONCLUSION

1)   Parts of this section repeat the recommendations (i.e. lines 380-386)! Need to be more focused on the conclusions! 

Our response: We have revised the conclusion to focus more on the findings of the study. See below:

The prevalence of caregiving has grown, and men are increasingly assuming roles as family caregivers. This scoping review examined racial and ethnic disparities in chronic stress, coping styles and health outcomes between white and African American male caregivers. A small number of studies showed that male African American male caregivers were economically disadvantaged financially, provided more hours of care, assisted with more ADLS and IADLS, were less religious, and were at higher risk for stroke than male white caregivers. Results showed a lack of studies on health disparities between white and African American caregivers. Recommendations were to adapt an intersectionality framework to study the cumulative effects of the inequities associated with demographic characteristics on caregiver burden and health outcomes and to develop interventions tailored to needs of these intersectional groups. Additional recommendations were to incorporate male perspectives and needs in research studies and in the design of interventions.

We hope that our responses satisfy your requirements. Once again, we thank you for your insightful comments and for the opportunity to revise and resubmit our manuscript.

Sincerely,

Christine Unson PhD

Anuli Njoku PhD

Stanley Bernard PhD

Martin Agbalenyo MPH

Round 2

Reviewer 1 Report

The authors did a great job responding to comments. There are some typos and minor grammar issues throughout that would require one read with a keen eye and then I think it's good.

Author Response

As requested, we engaged a professional to edit the manuscript.

All the edits are evident in the manuscript with the track changes.

Reviewer 2 Report

Dear authors,

I had the opportunity to read your revised version of the above manuscript. I realized that you have addressed in proper way almost all the issues I raised in my initial review

I think, the revised manuscript still needs an editing from a native English person.

Please see below some minor additional comments on specific points of the revised manuscript.

 1.      Introduction

lines 31-45: Please check the reference numbers Reference [1] is missing and since ref [2] is repeated, it creates confusion.

line 130: replace “The results of the study…” with “The results of the review…”

 2.      Literature Review/Theoretical Framework

line 64: Please add Thomson [17]…..

 3.      Results

Lines 188-189: “…five were published in the last 5 years [42-46, 50]…” You report six studies! Study # 44 was published in 2001. Please change the numbering

Table 1. It is not clear what exactly is described in column titled “Stress or Burden”

Although you replied that “We relabeled column heading titled “Stress or Burden” to “Findings about Caregiver Strain or Burden.”, the title has not changed in the revised version!

Author Response

Dear Sir or Madam,

We thank you for accepting our responses to your meticulous initial review of our manuscript. We are very appreciative of the time and effort you gave to help us improve its quality.

We have revised the manuscript according to your comments:

  1. Introduction

lines 31-45: Please check the reference numbers Reference [1] is missing and since ref [2] is repeated, it creates confusion.

Our response:  We have revised accordingly. Thank you for catching this mistake.

line 130: replace “The results of the study…” with “The results of the review…”

Our response: We have revised accordingly in several areas:

Introduction: Lines 129, 131

Discussion: Line 283

Recommendations: Line 314

  1. Literature Review/Theoretical Framework

line 64: Please add Thompson [17]…..

Our response: We have inserted as you noted (see line 64).

  1. Results

Lines 188-189: “…five were published in the last 5 years [42-46, 50]…” You report six studies! Study # 44 was published in 2001. Please change the numbering

Our response: We have made the change (see line 189).

Table 1. It is not clear what exactly is described in column titled “Stress or Burden”

Although you replied that “We relabeled column heading titled “Stress or Burden” to “Findings about Caregiver Strain or Burden.”, the title has not changed in the revised version!

Our response: We change the phrase to “Caregiver Stress or Burden Experienced” and added clarity by changing the table title to: Findings Related to Caregiver Stress or Burden, Coping Styles and Health Outcomes of Studies that Met Inclusion Criteria

Many thanks once again.